# Treatment of unresectable locally advanced pancreatic cancer with percutaneous irreversible electroporation (IRE) following initial systemic chemotherapy (LAP-PIE) trial: study protocol for a feasibility randomised controlled trial

Zainab L Rai ,[1,2] Veronica Ranieri ,[3,4] Daniel H Palmer,[5] Peter Littler,[6] Pauleh Ghaneh,[5] Kurinchi Gurusamy ,[1] Derek Manas,[7] Elena Pizzo,[8] Eftychia Eirini Psarelli,[9] Roopinder Gilmore,[1,10] Praveen Peddu,[11] David C Bartlett ,[12] Nicola de Liguori Carino,[13] Brian R Davidson[14]

For numbered affiliations see end of article.

**Correspondence to**
Dr Zainab L Rai;
zainab.rai.19@ucl.ac.uk

## ABSTRACT

**Background** Approximately 30% of patients with pancreas cancer have unresectable locally advanced disease, which is currently treated with systemic chemotherapy. A new treatment option of irreversible electroporation (IRE) has been investigated for these patients since 2005. Cohort studies suggest that IRE confers a survival advantage, but with associated, procedure-related complications. Selection bias may account for improved survival and there have been no prospective randomised trials evaluating the harms and benefits of therapy. The aim of this trial is to evaluate the feasibility of a randomised comparison of IRE therapy with chemotherapy versus chemotherapy alone in patients with locally advanced pancreatic cancer (LAPC).

**Methods and analysis** Eligible patients with LAPC who have undergone first-line 5-FluoroUracil, Leucovorin, Irinotecan and Oxaliplatin chemotherapy will be randomised to receive either a single session of IRE followed by (if indicated) further chemotherapy or to chemotherapy alone (standard of care). Fifty patients from up to seven specialist pancreas centres in the UK will be recruited over a period of 15 months. Trial follow-up will be 12 months. The primary outcome measure is ability to recruit. Secondary objectives include practicality and technical success of treatment, acceptability of treatment to patients and clinicians and safety of treatment. A qualitative study has been incorporated to evaluate the patient and clinician perspective of the locally advanced pancreatic cancer with percutaneous irreversible electroporation trial. It is likely that the data obtained will guide the structure, the primary outcome measure, the power and the duration of a subsequent multicentre randomised controlled trial aimed at establishing the clinical efficiency of pancreas IRE therapy. Indicative procedure-related costings will be collected in this feasibility trial, which will inform the cost evaluation in the subsequent study on efficiency.

**Ethics and dissemination** The protocol has received approval by London-Brent Research Ethics Committee reference number 21/LO/0077.
Results will be analysed following completion of trial recruitment and follow-up. Results will be presented to international conferences with an interest in oncology, hepatopancreaticobiliary surgery and interventional radiology and be published in a peer-reviewed journal.

**Trial registration number** ISRCTN14986389.

## STRENGTHS AND LIMITATIONS OF THIS STUDY

⇒ This trial is the first randomised comparison of irreversible electroporation therapy with 5-FluoroUracil, Leucovorin, Irinotecan and Oxaliplatin (FOLFIRINOX) vs FOLFIRINOX alone in patients with locally advanced pancreatic cancer (LAPC) and will aim to evaluate the feasibility of recruitment of patients with LAPC across multiple pancreas cancer specialist units in the UK.

⇒ As a feasibility study, it is not powered to show effectiveness; however, it is likely that the data will help inform a larger multicentre trial.

⇒ Experience in optimising recruitment, site setup and developing local radiology expertise will be crucial for the success of the subsequent trial, which will be powered from data collected during this feasibility study.

⇒ A potential limitation of the study is inadequate recruitment due to competing trials recruiting patients with LAPC.

## INTRODUCTION
### Pancreas cancer and LAPC
Pancreatic cancer (PC) is one of the deadliest cancers worldwide. It has a 5-year survival rate of less than 6% and an increasing incidence.[1] The low survival rate is mainly due to late diagnosis, as PC often remains asymptomatic until it becomes advanced. Surgery (pancreatic resection) is the only potentially curative treatment. However, only 10%–20% of patients are eligible for surgery.[2] Approximately 30% of all patients have locally advanced pancreatic cancer (LAPC) at diagnosis and are not considered for surgical resection.[3]

### Chemotherapy for LAPC
Systemic chemotherapy is the standard treatment for unresectable PC.[4] Despite advances in drug therapy for pancreas cancer over the last 20 years, the overall survival (OS) and progression-free survival (PFS) of these patients have not significantly improved.[5] Furthermore, a meta-analysis reported that chemoradiation is not superior to chemotherapy alone in patients with LAPC, with comparable OS and higher rates of grade 3–4 adverse events (AEs).[6]

In the UK National Institute of Clinical Excellence (NICE) guidelines, published in 2018, recommend systemic combination chemotherapy for both locally unresectable and metastatic PC.[7] NICE recommends first-line combination chemotherapy with 5-FluoroUracil, Leucovorin, Irinotecan and Oxaliplatin (FOLFIRINOX) in fit patients (Eastern Cooperative Oncology Group performance status of 0–1). In patients with comorbidities and restricted activities, or those unable to tolerate this regime, NICE recommends gemcitabine alone or gemcitabine-based therapy.[6] Although FOLFIRINOX shows the best objective response in PC,[8 9] it is associated with more frequent and severe side effects, including neutropenia, diarrhoea and peripheral neuropathy.[10]

### Repeat staging after first-line chemotherapy and possibility of resection
Staging is repeated following first-line chemotherapy for LAPC and imaging is reviewed once again regarding suitability for resection. In patients with locally advanced disease undergoing first-line treatment with FOLFIRINOX, 25.9% underwent resection, with an R0 resection rate of 78.4%; however, the long-term outcomes for these patients are not reported in the studies, and there is still a possibility of recurrence although at later date.[11]

### IRE for LAPC
Irreversible electroporation (IRE) is a novel ablative method for treating solid cancers.[12] IRE treatment destroys cancer cells using electricity. It can be delivered via an open approach involving a laparotomy and placement of probes around the cancer or via a percutaneous approach, which requires image guidance, via CT or ultrasound. Both methods require the patient to be under general anaesthesia with neuromuscular blockade.

A single study using propensity score matching (PSM) reported a median OS of 21.6 months in patients with LAPC treated with induction chemotherapy followed by IRE.[13] A systematic review on pancreas IRE therapy from 2019 reported an OS from diagnosis or treatment of up to 27 months and a morbidity rate of 30%.[14]

### First-line, up-front chemotherapy
The timing of chemotherapy in relation to IRE therapy may influence outcome. Studies where IRE has been delivered before chemotherapy have reported only modest increases in median OS,[15 16] whereas a systematic review evaluating the impact of IRE after induction treatment reported an OS of 27 months.[14] This may be because chemotherapy responders who have better outcomes are being selected to undergo IRE or alternatively it may suggest that chemotherapy modifies the tumour microenvironment making it more sensitive to IRE.

### Second-line chemotherapy for LAPC
NICE recommends that patients who have previously undergone FOLFIRINOX therapy as first line should be offered Gemcitabine-based second-line chemotherapy if their cancer has progressed on FOLFIRIONOX.[17]

### Locally advanced and borderline unresectable LAPC
The definition of borderline resectable disease is controversial,[18] whereas the definition of unresectable locally advanced disease has clear consensus, namely, the involvement of the coeliac artery or encasement of the superior mesenteric artery of more than 180° and or the involvement of the superior mesenteric vein or portal vein with no distant metastasis.[19] Therefore, the locally advanced pancreatic cancer with percutaneous irreversible electroporation (LAP-PIE) trial will only involve patients with locally advanced disease fulfilling published criteria.

### Complications of pancreas IRE therapy
Reported complications following pancreas IRE include pancreatic fistula, venous and arterial thrombosis, arterial pseudoaneurysm and pancreatic abscess.[14] The mean morbidity rate in the studies examined is 34% but can reach up to 62%.[20] The average rate of severe complications (defined as greater than grade III on the Clavien-Dindo scale) following IRE is 8% but can be as high as 44%.[21] The average procedure of related mortality rate in these studies is 3%.[12]

There are significant variations in the percentage and severity of complications reported. This difference may be due to the case selection, whether IRE is carried out during surgery (via an open or laparoscopic approach) or with image guidance in the interventional radiology (IR) suite, the treatment algorithm, the completeness of reporting from treatment centres or the experience of the radiologist and/or centre. Only two studies specifically mention the experience of clinicians delivering IRE[16 22] and, currently, there is limited published data available exploring the number of IRE procedures performed and the relationship with outcomes.[23]

IRE is a palliative option and, therefore, its impact on patient's quality of life (QoL) is an essential consideration. A single-centre prospective study evaluating the QoL pre and post-IRE in patients with LAPC demonstrated that IRE had no adverse effect on QoL in the short-term; however, concomitant treatment with chemotherapy after IRE as well as other inter-related factors may confound the results.[24] We were unable to find any studies comparing QoL changes following IRE to a control group. The published data, to date, regarding the improvement in QoL in patients with LAPC undergoing IRE is weak and needs further exploration.

The current information available in the literature regarding the use of IRE in the treatment of LAPC is primarily from retrospective and prospective cohort studies. A single prospective randomised controlled trial (RCT) has recently reported on the use of IRE in LAPC with concurrent gemcitabine therapy.[25] The authors report a survival advantage in patients treated with IRE and gemcitabine. Therefore, some evidence is available for the efficacy of IRE in LAPC; however, the LAP-PIE trial will evaluate whether this is also the case in the modern context of FOLFIRINOX chemotherapy, a regimen recommended in the UK by NICE. The LAP-PIE trial is a feasibility trial to determine whether patients would be willing to have pancreas IRE treatment, whether they would accept randomisation to IRE or standard of care chemotherapy, whether they would comply with treatment and study data collection and to collect preliminary data on efficacy end points and indicative costs related to health resources, with which to design a subsequent trial on treatment efficiency and cost-effectiveness.

## METHODS AND ANALYSIS
### Study design and setting
The LAP-PIE trial is a UK multicentre feasibility study that will recruit 50 patients with LAPC in whom the disease has remained localised and unresectable following FOLFIRINOX. Patients will receive 3–6 months of FOLFIRINOX prior to restaging and being assessed for study eligibility. All participants will be offered palliative chemotherapy according to standard centre protocols and half will also be offered one IRE procedure. If new agents for second-line therapy are approved by NICE prior to or during the LAP-PIE trial, these can be offered in the second-line chemotherapy arm and following IRE therapy if considered appropriate by the treating oncologist. Patients randomised to the control arm who have benefitted from initial chemotherapy with FOLFIRINOX may continue on the same regimen if this deemed appropriate by the treating oncologist. It is an open-label study. Patient enrolment is expected to last 15 months assuming a recruitment rate of 0.8 patients per site per month across seven UK centres and an opening site rate of 0.5 per month. The trial will be conducted in specialist pancreas cancer centres. IRE will be delivered across all trial centres. Interventional radiologists performing

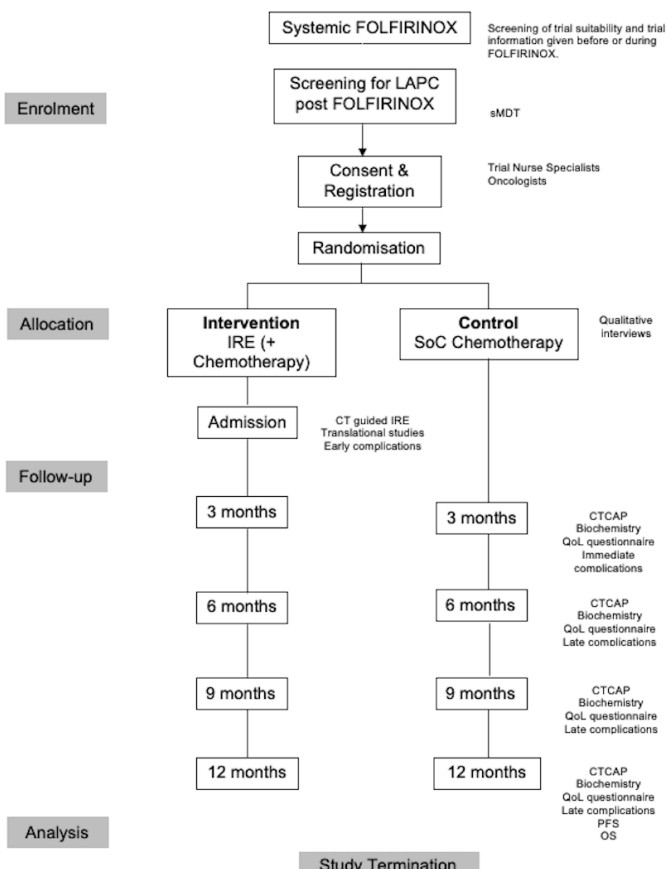

**Figure 1** Trial flow diagram. CTCAP, CT chest abdomen pelvis; FOLFIRINOX, 5-FluoroUracil, Leucovorin, Irinotecan and Oxaliplatin; IRE, irreversible electroporation; LAPC, Locally Advanced Pancreatic Cancer; OS, overall survival; PFS; progression free survival; QoL, quality of life; SoC, standard of care; sMDT, specialist multidisciplinary team.

pancreas IRE in the trial will have performed a minimum of five previous pancreas or liver IRE ablations prior to administrating IRE to participants enrolled in the trial. Additional technical support will be available from manufacturers of the equipment, AngioDynamics . Figure 1 summarises the trial design and table 1 demonstrates the schedule of enrolment, interventions and assessments.

### Participants
Adults (≥18 years) with LAPC following first-line chemotherapy with FOLFIRINOX will be eligible. Only patients who have completed FOLFIRINOX as their first-line chemotherapy will be eligible. Patients who have received FOLFIRINOX as second-line therapy prior to inclusion will not be eligible. Following FOLFIRINOX chemotherapy, patients will be restaged with CT chest, abdomen and pelvis (CAP)±an *fluorodeoxyglucose* (FDG) positron emission tomography (PET) scan, depending on local site protocols. The imaging will be reviewed in a specialist multidisciplinary team (sMDT) meeting.

Patients with borderline resectable disease will be excluded. The definition for this is summarised in table 2. Other exclusion criteria include untreated biliary or gastric outlet obstruction, coagulopathy, chronic kidney

**Table 1** Schedule of enrolment, interventions and assessments of the LAP-PIE trial

| | Study period | | | | | | | | | |
|---|---|---|---|---|---|---|---|---|---|---|
| | Enrolment | Allocation | Post-allocation follow-up time points | | | | | | | Close-out |
| Timepoint | $-t_1$ | 0 | Pre-IRE | 6hours post-IRE | 6days post-IRE | 3 months | 6 months | 9 months | 12 months | |
| **Enrolment:** | | | | | | | | | | |
| Eligibility screen | X | | | | | | | | | |
| PIS given | X | | | | | | | | | |
| Systemic treatment with FOLFIRINOX | X | | | | | | | | | |
| sMDT | X | | | | | | | | | |
| Informed consent | X | | | | | | | | | |
| Allocation | | X | | | | | | | | |
| **Interventions:** | | | | | | | | | | |
| IRE (intervention) | | | ←————————→ | | | | | | | |
| Systemic chemotherapy (SoC) | | | | | | ←————————→ | | | | |
| **Assessments:** | | | | | | | | | | |
| CT CAP | X | | | | | X | X | X | X | |
| Blood tests | X | | X | X | X | X | X | X | X | |
| HRQoL | | | | | | X | X | X | X | |
| Complications | | | | | | X | X | X | X | |

CT CAP, CT chest abdomen pelvis; FOLFIRINOX, 5-FluoroUracil, Leucovorin, Irinotecan and Oxaliplatin; HRQoL, health-related quality of life; IRE, irreversible electroporation; LAP-PIE, locally advanced pancreatic cancer with percutaneous irreversible electroporation; PIS, patient information sheet; sMDT, specialist multidisciplinary team; SoC, standard of care.

**Table 2** Definition of borderline resectable pancreatic cancer[18]

|  | Borderline resectable | Unresectable |
|---|---|---|
| Coeliac artery | Tumour without encasement or abutment | Tumour contact >180° |
| Superior mesenteric artery | Tumour abutment ≤180° | Tumour contact >180° |
| Superior mesenteric vein (SMV)/portal vein (PV) | Tumour abutment of SMV/PV>180° or abutting ≤180° with irregularity of the vein ±thrombosis with anatomical structures that still permit safe and complete resection with vein reconstruction | SMV/PV unreconstructible secondary to tumour involvement or occlusion |
| Common hepatic artery | Reconstructible short segment abutment | Unreconstructible interface between tumour mass and vessel |

Adapted from Callery *et al*.[18]

disease (CKD) stage ≥3 (the criteria for CKD stage 3 is an estimated glomerular filtration rate of between 30 and 59),[26] indwelling electrical devices, abdominal varices precluding safe access to the pancreas and significant comorbidities that would contraindicate general anaesthesia and neuromuscular blockade. A contraindication, as per manufacturer guidance, is epilepsy as the electrical discharges from IRE treatment may in theory result in seizure activity through pulsatile brain stimulation. Although the overall impact of IRE on seizure activity and neurological conditions remains unclear,[27] in the interest of robust patient safety, patients with neurological conditions will be excluded. The full inclusion and exclusion criteria are summarised in figure 2.

### Randomisation
Following confirmation of eligibility randomisation will be performed by the Liverpool Clinical Trials Centre (LCTC) using a web-based randomisation procedure. The LCTC will inform local centres of the treatment arm and the local LAP-PIE Clinical Trials Nurse will inform the patient and make arrangements for clinical review and the date for the chosen therapy.

**Inclusion criteria**
- Tissue confirmation of adenocarcinoma
- Successful completion of FOLFIRINOX
- Cancer deemed unresectable at sMDT review
- Ability to provide informed consent
- ≥ 18 years of age
- Considered amenable to IRE by pancreas IR
- WHO Performance status 0 /1.
- Maximum cancer diameter of 3.5 cm at time of IRE treatment.
- Medically fit for general anaesthetic

**Exclusion Criteria**
- First line chemotherapy other than FOLFIRINOX
- Concomitant malignancy (except skin and prostate cancers)
- Metastatic/Borderline resectable disease
- Untreated gastric outlet/biliary obstruction
- Co-morbidity precluding general anaesthesia +/- neuromuscular blockade
- Indwelling electrical devices
- CKD≥3
- History of neurological disease
- Abdominal varices
- Active bleeding
- Anticoagulation which cannot be discontinued,
- Coagulopathy (INR>1.5)/**any one** of the following:
  o Hb< 8 g/dL
  o absolute neutrophil count< 1500 cells/mL
  o platelet count< 50,000

**Figure 2** Inclusion and exclusion criteria of the LAP-PIE trial. CKD, chronic kidney disease; FOLFIRINOX, 5-FluoroUracil Leucovorin, Irinotecan and Oxaliplatin; Hb, haemoglobin; IR; interventional radiologist; IRE, irreversible electroporation; LAP-PIE, locally advanced pancreatic cancer with percutaneous irreversible electroporation; sMDT, specialist multidisciplinary team.

### Intervention
#### Pancreas IRE
Patients will be invited for an anaesthetic preassessment to ensure that they are fit enough to undergo IRE under general anaesthesia prior to being booked for hospital admission. Patients will be admitted to a ward or radiology admissions unit under joint clinical care of a pancreas interventional radiologist and one of the other site leads (surgery, Oncology).

The CT CAP performed on completion of FOLFIRINOX will be used by the interventional radiologist to plan IRE therapy. Under general anaesthesia, IRE will be delivered percutaneously using CT guidance. Timing of the procedure will be coordinated with the treating oncologist to ensure bone marrow recovery following FOLFIRINOX chemotherapy. A broad-spectrum antibiotic, based on individual centres' first-line prophylaxis for Gastro-Intestinal (GI) infection, will be given prior to the procedure and 6 hours and 12 hours following IRE.

The IRE procedure will be standardised to use between 2 and 6 IRE needles depending on the size, shape and relationship of critical structures to the tumour, with a maximum interelectrode distance of 2 cm. A treatment margin of at least 0.5 cm will be maintained. The needle placement prior to treatment will be recorded on CT.

IRE therapy will be delivered using ECG synchronisation to ensure that the pulse is delivered during diastole. The length of the electrical pulse and amperage will be decided by the treating interventional radiologist and will be between 70 μs and 100 μs with an amperage of between 20 AMPs and 40 AMPs. The treating interventional radiologist will have relevant expertise in IRE procedures with a minimum of five IRE procedures on liver or pancreas performed previously.

Following IRE therapy, full blood count, liver function tests, coagulation, renal function and amylase (or lipase) will be monitored and patients will be admitted for a minimum of 24 hours to ensure adequate pain management and to monitor and manage any immediate complications. Participants will be discharged with pain relief as required and Deep Vein Thrombosis (DVT) prophylaxis for 4 weeks.

The intraoperative imaging will be reviewed by the lead radiologist at each site and if the entire tumour has been treated based on these images, the procedure will be deemed as successful. The technical success will be confirmed by central review. Contrast CT follow-up scans will be assessed for evidence of tumour growth as indicated by mass enlargement or increased vascularity. In the absence of tumour growth, the procedure will be considered successful at follow-up. At the end of the follow-up period, serial imaging will be analysed, and the tumour staged. Local disease control will be evaluated according to the Response Evaluation Criteria in Solid Tumours. This will be performed by the radiology leads at each site and confirmed by a central review.

Following IRE, patients in this arm will be offered second-line chemotherapy, which is the current standard of care. The drug choice and timing of chemotherapy will be decided by treating oncologists and will adhere to NICE guidelines.

### Controls (second-line systemic chemotherapy)

Patients randomised to receive further chemotherapy (chemotherapy alone) will commence treatment within 6 weeks after sMDT or 4 weeks after randomisation, whichever is sooner. The second-line chemotherapy regime will be selected at the discretion of the treating oncologist and will adhere to NICE guidelines and local protocols.

### Outcomes

The primary objective of the LAP-PIE trial is to ascertain feasibility of such a trial concept. In particular, the ability to identify and recruit LAPC patients at the selected sites to the LAP-PIE trial will be evaluated. The primary outcome consists of recruitment rate, number of screening failures, number of patients completing study pathway as per protocol and trial withdrawal rate.

The secondary objectives of the LAP-PIE trial aim to assess the practicality and technical success rate of pancreas IRE treatment. Technical success will be determined by the interventional radiologist performing the procedure. The procedure will be deemed to be technically successful if the entire tumour has been treated based on imaging performed during the intervention. Completion and complication rates of IRE procedures will be evaluated. The proportion of patients in whom percutaneous IRE was successful, defined as no local growth on follow-up, will be determined as will the proportion of patients who became eligible for surgery following IRE and the R0/R1 resection rates in these participants. Acceptability of pancreas IRE therapy to patients and their clinicians, rate of withdrawal from the study and outcome measures including OS (measured from randomisation), PFS and QoL will be evaluated. Finally, health-related costs, including indicative costs related to health resource use in both arms, social costs of attending IRE treatment such as cost of travel to and from a tertiary centre, time off work and social support costs will be evaluated.

### Follow-up

All patients will be clinically reviewed every 3 months during which a CT CAP will be performed. Additional imaging can be performed if clinically indicated. Other investigations will include biochemistry and haematology, CA19–9 levels, C Reactive Protein (CRP), QoL as measured by a validated questionnaire (EQ5D-5L)[28] and a clinical review of any AEs experienced. This information will be recorded in an electronic case report form.

### Sample size calculation

Formal sample size calculations are not appropriate for a feasibility study. Sample sizes between 24 and 50 have been recommended to estimate the SD required for a sample size calculation to allow for the design of a subsequent RCT aimed at evaluating the cost-effectiveness of IRE in patients with LAPC.[29 30] A convenient sample size of 50 patients, randomised equally to the two groups, has been selected.

### Patient and public involvement

Patient and public involvement representatives have been involved in designing the trial ensuring that the patient interests are central to the study. The design of the trial was presented to the Pancreas Cancer Action Day, which has a large representation from patients and carers and received extremely positive feedback. Two patient representatives will serve on the trial management group and will be involved in informing, monitoring and helping to interpret the findings from the study. Both patient representatives were involved in the trial design and in developing the trial protocol and patient information sheets. As the trial progresses, they will help address any patient-related concerns and will have input into the interpretation of the trial outcomes and how this will be communicated to the patient user groups. They will then be involved, along with other patient advocates, the trial management team and other key invited participants in the discussion of the trial findings at a focus group meeting and the preparation of the protocol for a subsequent trial addressing efficacy and cost-effectiveness of IRE therapy in pancreas cancer. Once the data analysis and interpretation has been completed, they will contribute to the dissemination of the trial outcome information to patients' support groups via social media.

### Assessment of feasibility

One aspect of the feasibility evaluation of this trial is to evaluate the ability to recruit patients with LAPC into a trial of this nature. The average number of patients with unresectable LAPC within UK specialist pancreas units is 60 patients per centre per year (Royal Free Hospital London audit data 2017, unpublished). Staging following FOLFIRINOX chemotherapy will demonstrate localised but unresectable disease in approximately 40 out of 60 patients (65%).[12] These patients will be eligible for the LAP-PIE trial. We anticipate recruiting 20% of potential participants to the LAP-PIE

trial, which equates to eight per centre per year. Recruitment within five centres would allow 40 UK patients per year. A recruitment time of 15 months would allow 50 patients to be recruited.

## Assessment of complications

AEs for the Standard of Care Chemotherapy will be defined using the Common Terminology Criteria for Adverse Events (CTCAE) V.4.0.[31] AEs for IRE will be defined using the Clavien Dindo classification of surgical complications.[32] Complications greater than grade 3 on the CTCAE scale and greater than grade III on the Clavien Dindo scale will be reported.

A serious AE for patients randomised to receive standard of care chemotherapy only will be compared against the reference safety information (RSI) within the current summary of product characteristics (SmPC) for that drug. The RSI will be taken from the latest SmPC available for the drug on https://www.medicines.org.uk/emc. The RSI for IRE will be from the Nanoknife system users' manual.

Suspected unexpected serious adverse reaction (SUSAR) will be defined as life threatening, serious and non-recognised complications. Two or more reports of SUSARs related to IRE therapy will result in the cessation of trial activity in all centres to allow for a thorough investigation into the safety of IRE. Any premature halt in the trial will be reported to the Medicines and Healthcare products Regulatory Agency and the Regional Ethics Committee (REC).

## Qualitative study in LAP-PIE trial

A qualitative study will be carried out to understand patients' and clinicians' perspectives and experiences of the procedure and the design of the trial. All patients approached to participate in the trial will be invited to participate. A purposive sample of healthcare professionals involved in the trial will also be invited. Trial nurses, local R&D (research and development) personnel and Network CRN (clinical research network) managers will be included. This information will be gathered via interviews by an experienced qualitative researcher.

Interviews will be transcribed verbatim and managed using computer-assisted qualitative data analysis software (iNVivo).[33] Emerging data will be used to modify trial information and protocol. This may include the development of training materials for staff and changes to participant information sheets.

## Translational study

A translational substudy will evaluate prognostic markers for outcome. Imaging prior to and during IRE therapy will be correlated with outcome as will the immune response to IRE therapy by assessing circulating immune responses (including absolute lymphocyte count, eosinophils, neutrophil: leucocyte ratio, Tregs and IFNγ at baseline, 6 hours and 6 days post-therapy.

## Progress to full trial

The criteria to progress to a subsequent full trial will be determined quantitatively as (A) the ability to recruit 50 patients to the study across the study sites (B) a consent rate of 20±11%, (C) overall technical success rate of 50±14% and (D) a loss to follow-up of 10±8%.

## Ethics

The trial will abide by the principles of the World Medical Association Declaration of Helsinki and has approval from London Brent Research Ethics Committee, reference number 21/LO/0077. The trial has been registered on International Standard Randomised Controlled Trial Number https://doi.org/10.1186/ISRCTN14986389.

## Publication and dissemination

Results from different centres will be analysed and published together following completion of the recruitment and follow-up period. This will be presented to an international conference with an interest in hepatopancreaticobiliary surgery, oncology and IR and published in a peer-reviewed journal.

# DISCUSSION

## Trial justification

Local ablative therapy using IRE in patients with LAPC has been shown to be beneficial in prospective and retrospective studies (median survival 27 months);[14] however, the increase in survival reported may be due to patient selection for IRE rather than the effect of IRE (selection bias). A systematic review of IRE therapy in LAPC included retrospective and prospective studies but no RCTs.[14 34] Moreover, the heterogeneity among individual trial participants including varying concomitant treatments prevented combination of data sets making it difficult to establish the true impact of IRE. Despite this, the systematic review is a useful aid in summarising findings including complication rates following IRE and highlighting the current knowledge gaps such as the relationship between number of procedures performed and outcome and the impact of IRE on QoL.

A study using PSM reported a median OS of patients with LAPC undergoing IRE of 21 months,[13] similar to that reported by the systematic review (27 months).[14] Such a study may effectively adjust for observed variables between cohorts and can contribute to a more precise estimation of treatment response. However, selection bias cannot be completely eliminated because PSM does not adjust for unobserved differences between groups. In order to address selection bias, a well-designed randomised control trial is required.

Before IRE can be justified by NICE to be incorporated into routine clinical practice, a randomised control trial is needed evaluating the benefit of IRE in addition to standard of care chemotherapy, which in the UK is FOLFIRINOX. While a larger trial is required to investigate IRE treatment in LAPC, patients and clinicians may not be

willing to enter the trial and, therefore currently a large RCT is justified but not feasible.

## Importance of doing a feasibility study

Clinical trials comparing different treatment types have high rates of failure.[35] This is particularly true where there has been a lack of feasibility testing and qualitative evaluation of the study design.[26 27] Furthermore, if a feasibility study demonstrates that a full trial would be unlikely to meet the required patient recruitment rate, intervention acceptability and other feasibility outcomes, then the potential wasted cost of a larger clinical trial is avoided making feasibility trials an important first step.[36] A feasibility trial will enable a realistic assessment of the capability of registered sites in patient enrolment and will compare IRE to the current standard of care. The qualitative study will explore the patient evaluation of information provided, the quality and clarity of this information, the individual providing the information, the clinical equipoise, support and attitude of other clinicians and general practitioners and the views and experiences of patients' family and friends. The feasibility trial will provide information about challenges that may be encountered during enrolment and these factors can be mitigated for if a larger RCT is justified and subsequently planned.

## Exclusion of borderline resectable disease

Patients with borderline resectable disease have been excluded from the LAP-PIE trial. The exact definition of borderline resectable pancreatic disease (BRPD) varies, but broadly it is characterised as tumour abutting or involving vascular structures.[18] A number of studies have suggested that neoadjuvant chemotherapy or chemoradiotherapy may downsize the tumour allowing resection and preventing local recurrence.[37–39] The LAP-PIE trial will exclude patients with BRPD because of the variability in radiological criteria used for BRPD and to avoid patients being recruited to the trial who would be considered in some pancreas units for surgical intervention.

## Choice of FOLFIRINOX as first line

Previous studies have reported no significant improvement in survival when IRE is used as a first-line treatment for unresectable LAPC.[15 16] Furthermore, systemic chemotherapy has previously been shown to downgrade a proportion of locally advanced unresectable cancer into resectable disease and, therefore, delivering IRE first line may result in an unnecessary, invasive procedure in a group of patients whose tumours may be downgraded and who would go on to have potentially curative resectional surgery.

FOLFIRINOX was specifically selected as the regime of choice because it has been shown to result in the greatest number of patients downgraded to resectable disease.[10] Therefore, the patients recruited into the LAP-PIE trial will be those with truly unresectable disease in whom the impact of IRE can be evaluated.

## Translational studies

The translational study will determine the radiological and immune response of patients undergoing IRE and determine its biological significance with emphasis on OS and disease response. This data will provide valuable information about the type of patient that is likely to benefit the most from IRE and, therefore, form the basis of a management design that is tailor made to individual patients.

Mandated imaging in the LAP-PIE trial is a CT scan and this will provide information about contrast enhancement features. The images will be reviewed and information about tumour characteristics such as size, histology and grade will be collected. The relationship between treatment parameters and imaging findings during the IRE procedure, as well as pretreatment PET scans, will be analysed. The data generated will be used to develop a mathematical model to predict IRE responses and help understand the significance of radiological changes and their biological significance. Blood samples will be taken from patients enrolled in the study and analysed for peripheral immune response markers. This will include absolute lymphocyte count and eosinophils, neutrophil: leucocyte ratio, Tregs and IFNγ. Previous studies have demonstrated a detectable peripheral immune response in patients following IRE.[40 41] Analysing the immune response of patients undergoing IRE will provide information about whether a peripheral immune response can be determined by simple biochemical tests in these patients and if this can predict treatment efficacy. This information may be used in the future to predict treatment response and guide future patient selection.

The qualitative substudy will provide the opportunity to collate and analyse the experiences of both patients and clinicians in order to understand the perspectives and experiences of the procedure and trial providing useful information on the barriers and facilitators experienced by both groups.

Preliminary studies have shown that IRE in addition to chemotherapy may provide survival benefit in patients with LAPC.[14–16 21–23 34] However, this may be due to patient selection for IRE. There are no RCTs addressing the issue. The LAP-PIE trial is the first UK-based RCT of pancreas IRE in patients with LAPC. The trial will assess feasibility and safety of delivering this intervention and if successful, it will progress to a subsequent large-scale RCT trial in assessing the efficacy and cost-effectiveness of IRE therapy in PC. This research is required to investigate whether IRE improves survival, health-related QoL or both.

**Author affiliations**
¹Division of Surgery and Interventional Science, University College London, London, UK
²Wellcome/EPSRC Centre for Interventional and Surgical Sciences (WEISS), London, UK
³Research & Development, Tavistock and Portman NHS Foundation Trust, London, UK
⁴Science & Technology Studies, University College London, London, UK

⁵Hepato-Pancreatco-Biliary Disease, University of Liverpool, Liverpool, UK
⁶Hepato-Pancreatico-Biliary Surgery, Freeman Hospital, Newcastle Upon Tyne, UK
⁷Hepato-Pancreatico-Biliary Surgery, Newcastle Upon Tyne Hospitals NHS Trust, Newcastle Upon Tyne, UK
⁸Applied Health Research, University College London, London, UK
⁹Liverpool Clinical Trials Centre, University of Liverpool, Liverpool, UK
¹⁰Oncology, Royal Free London NHS Foundation Trust, London, UK
¹¹Department of Radiology, King's College Hospital NHS Trust, London, UK
¹²Hepato-Pancreatico-Biliary Surgery, University Hospitals Birmingham NHS Foundation Trust, Birmingham, UK
¹³Regional Hepato-Pancreato-Biliary Unit, Manchester Royal Infirmary, Manchester, UK
¹⁴HPB Surgery, Royal Free London NHS Foundation Trust, London, UK

**Acknowledgements** This project is funded by the National Institute for Health Research (NIHR) under its Research for Patient Benefit (RfPB) Programme (PB-PG-1217-20042). The views expressed are those of the author(s) and not necessarily those of the NIHR or the Department of Health and Social Care.

**Contributors** All listed authors meet the ICMJE criteria. We attest that all authors contributed significantly to the creation of this manuscript, each having fulfilled criteria as established by the ICMJE. We confirm that the manuscript has been read and approved by all named authors. We confirm that the order of authors listed in the manuscript has been approved by all named authors. ZLR: manuscript preparation. VR: trial management. DHP: protocol design, trial management and co-applicant. PL: protocol design, trial management and co-applicant. EEP: statistical analysis plan. KG: major contribution to design and funding application. DM: protocol design, trial management and co-applicant. PG: protocol design, trial management and co-applicant. EP: health economic analysis design. RG: protocol design, site lead, PP: protocol design, site lead, DCB: site lead, NLC: site lead. BRD: principal investigator and grant holder, overall trial management, study design and review of manuscript. All authors read and approved the final manuscript.

**Funding** The LAP-PIE trial is funded by the National Institute for Health Research (NIHR) under its Research for Patient Benefit (RfPB) Programme (PB-PG-1217-20042). EP is supported by the National Institute for Health Research ARC North Thames. The views expressed in this publication are those of the author(s) and not necessarily those of the National Institute for Health Research or the Department of Health and Social Care.

**Competing interests** None declared.

**Patient and public involvement** Patients and/or the public were involved in the design, or conduct, or reporting, or dissemination plans of this research. Refer to the Methods section for further details.

**Patient consent for publication** Not applicable.

**Provenance and peer review** Not commissioned; externally peer reviewed.

**ORCID iDs**
Zainab L Rai http://orcid.org/0000-0002-6799-0585
Veronica Ranieri http://orcid.org/0000-0003-0528-8640
Kurinchi Gurusamy http://orcid.org/0000-0002-0313-9134
David C Bartlett http://orcid.org/0000-0001-8564-9420

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
