## [Reviewer comments · BMJ Open]

ARTICLE DETAILS

TITLE (PROVISIONAL)	Treatment of unresectable Locally Advanced Pancreatic cancer with Percutaneous Irreversible Electroporation (IRE) following initial systemic chemotherapy (LAP-PIE) trial: Study protocol for a feasibility randomised controlled trial.
AUTHORS	Rai, Zainab; Ranieri, Veronica; Palmer, D. H.; Littler, Peter; Ghaneh, Pauleh; Gurusamy, Kurinchi; Manas, Derek; Pizzo, Elena; Psarelli, Eftychia; Gilmore, Roopinder; Peddu, Praveen; Bartlett, David; de Liguori Carino, Nicola; Davidson, Brian

VERSION 1 – REVIEW

REVIEWER	Flak, Rasmus Virenfeldt Aalborg Universitetshospital, Department of Gastrointestinal Surgery
REVIEW RETURNED	21-Apr-2021

GENERAL COMMENTS	Thank you for letting me review this study protocol. I have made several comments of general and specific nature regarding the manuscript. I have not checked whether the details of the manuscript accurately reflect the attached study protocol/SPECTRE checklist. General remarks: 1) The feasibility of percutaneous IRE in LAPC patients has been shown in multiple publications. Why would the authors go through the effort of designing and conducting a randomized controlled trial and not power it for efficacy?2) The introduction/background section is hard to read. The authors are encouraged to rethink the structure of this section. A structure could be to first explain the standard treatment program and then go on to explain how IRE has been implemented in the existing clinical trials. Finally, please explain why the implementation in the current study is chosen and what the aims of the study are.3) Are IRE treatments being performed in all centers or just one. Please elaborate on the prior experience with IRE treatment.4) How is safety maintained throughout the study? The protocol does not appear to contain any constraints or limitations on how many and how severe adverse event are acceptable to continue the trial. Please consider formulating a clear guideline for when the trial is considered too dangerous to continue. Specific remarks: Introduction/background 1) Page 8, line 51-59: The authors wrote that patients not eligible for FOLFIRINOX are treated with gemcitabine monotherapy and
---

then goes on to say that NICE guidelines recommend GemCap. This seems to be inconsistent. The authors are encouraged to rephrase or explain?

2) Page 9, line 11-12: The headline is not consistent with the content of the paragraph. Upfront chemotherapy, consolidative chemotherapy or downstaging chemotherapy are better descriptors than “First-line chemotherapy” in this context. Maybe “timing of IRE treatment” is a better headline.

3) Page 9, line 19-22: The authors wrote that “studies” report an OS of 27 months, but only reference one study. The authors are encouraged to rephrase.

4) Page 9, line 38-48: Considering that the cited study is a systematic review, the statement “and in this cohort” seem out of place. The authors are encouraged to rephrase.

5) Page 10, line 3-6: Please refer to the specific guidelines use to define LAPC. In this case I think it is the NCCN guideline, as these are referenced in the paragraph.

6) Page 10, line 8-27: This should be in the methods section.

7) Page 11, line 15: “during surgery or electively in the IR suite”. I believe that the authors meant to differentiate between open surgery (open in-situ IRE) and percutaneous image-guided IRE. Please make this clearer. Also, IR (interventional radiology?) is not explained.

8) Page 11, line 22-24: It is not completely true that no data exist regarding the number of treatments performed and outcomes. Please see: <Philips P, Hays D, Martin RCG (2013) Irreversible Electroporation Ablation (IRE) of Unresectable Soft Tissue Tumors: Learning Curve Evaluation in the First 150 Patients Treated. PLoS ONE 8(11): e76260. doi:10.1371/journal.pone.0076260>

9) Page 11, line 30-37: Please reread the article. The authors find that QoL is impacted in the short term, but suggest that this could be the results of other interrelated factors.

10) Page 11, line 36-39: What is meant by “no recognised QoL scores that compare IRE with no treatment or systemic chemotherapy”? I suggest rephrasing to “We were unable to find any studies comparing the QoL changes after IRE to a control group.”

11) Page 11, line 39-44: None of the published data, to date, has found any positive effects on QoL. What data are the authors referring to?

12) Page 11, line 46-52: This paragraph seems out of place and should be removed.

13) Page 11, line 57-13: Why demonstrate cost-effectiveness, when efficacy has not been established? Please explain. Also, if the aim of the trial is to assess “whether they would accept randomization to IRE or standard of care chemotherapy, whether they would comply with treatment and study data collection”, why not set up an even smaller trial to asses this prior to a larger randomized controlled trial? Considering that IRE has been linked to severe morbidity and mortality, please explain why 50 patients are needed to asses this!

Methods section:

1) Inclusion criteria: Tissue confirmation of what? Adenocarcinoma? Please be specific. What defines a “successful completion of FOLFIRINOX”? Please elaborate. Consider rephrasing inoperable to unresectable. Please explain “IR”.

2) Exclusion criteria: What if FOLFIRINOX has been given in the second-line but prior to inclusion? It is seldom, but can happen in

	some patients, because of positive changes in performance status after initial therapy. Again, please be precise in the use of the words first/second-line chemotherapy. Also, this criterion seems to be redundant as patients are only included if they have completed FOLFIRINOX. Please consider reading the literature on IRE ablations in the presence of metallic biliary stents and consider adding this as an exclusion criterion. 3) IRE-procedure: Consider standardizing the energy delivery. Why deviate substantially from the published literature? 4) Page 19, line 12-22: The authors have previously written that both groups will be offered additional chemotherapy (please consider calling it additional palliative chemotherapy or adjuvant chemotherapy). In this paragraph, the authors write that patients are randomized to receive chemotherapy. Please be consistent. Consider rephrasing to “chemotherapy only”. 5) Outcomes: Why is cost-effectiveness not mentioned here, when this was considered a primary aim of the trial in the background section? In the secondary outcomes, please be specific on what is measured, e.g. “...safety and complications of IRE treatment, technical success of intervention...” is not apparently measurable outcomes. Also, how is PFS “measured as a binary outcome”? Please explain. How are QoL evaluated? Are the authors interested in measuring changes in certain QoL outcomes or just means at specific points in the follow-up-period? 6) Sample size: Please read comment above. 7) Page 20, line 52: Interpretation of results should be objective! 8) Assessment of feasibility: The evaluated recruitment rate seems to be high, considering the number of patients and the selection criteria. The authors should ask themselves how many patients are still in PS 0/1 after FOLFIRINOX treatment, are unresectable and have tumors below 3.5cm in diameter. 9) Assessment of complications: Consider using the same classification system in both groups of patients. Otherwise, they will not be comparable in the analysis. Additionally, what is the point in comparing SAEs to the reference and not to the other group? Please explain why this is not compared between groups in a RCT. 10) Translational study: Please be specific about what exactly is measured. Ethics: 1) Please reference the specific REC. 2) Please register protocol in a public database e.g. clinicaltrials.gov. Discussion: 1) Page 24, line 10-15: Propensity score matched studies compare two groups of patients, while single-arm follow-up studies do not. Consider reporting the difference between the two groups instead of the OS of one group. 2) Page 25, line 56-3: The authors should consider the context of the treatment here. FOLFIRINOX has been shown to reduce tumors to a resectable size, even when this is not evident from the clinical images. Please read: “Ann Surg Oncol (2015) 22:1153–1159 DOI 10.1245/s10434-014-4225-1”. The authors are encouraged to consider how they will avoid including patients that are potentially resectable. 3) Page 26, line 40-43: This information should be mentioned in the methods section. Please explain when the blood samples are taken and how the comparison of the results will be performed.
--	---

	4) Page 26, line 47-52: “analyzing...efficacy”. What is meant by “defined” this sentence? Please rephrase. 5) Page 27, line 12-15: Please provide reference. 6) Page 27, line 38-41: “The LAP-PIE trial will provide evidence for supporting or refuting the role of IRE in patients with locally advanced pancreatic cancer.” How is this possible if the study is not powered to do so?
--	--

REVIEWER	Tang, Monica University of New South Wales, Centre for Big Data Research in Health
REVIEW RETURNED	26-Apr-2021

GENERAL COMMENTS	The file titled SEPTRE checklist appears to be a duplicate copy of the trial protocol. This protocol describes a feasibility study for a novel procedure for locally advanced pancreatic cancer, percutaneous irreversible electroporation. The study is not powered to demonstrate efficacy, but data from the study will form basis of an application for a larger trial. The protocol is well-designed, well-written and is an appropriate platform for investigating an innovative treatment. Major comments  1. This feasibility study is described as the pre-cursor to conducting a larger trial that will assess the cost-effectiveness of this procedure. If a feasibility trial is positive, typically the next step would be to conduct a randomised controlled trial to demonstrate efficacy. Cost-effectiveness becomes a relevant question only after efficacy is demonstrated in a randomised trial, so it unclear why a cost-effectiveness study would directly follow a positive feasibility trial. 2. More details should be provided regarding the systemic treatment components of the study, as they are likely to affect tolerability of the overall protocol and potentially survival outcomes. How many cycles of FOLFIRINOX will patients receive prior to restaging and being assessed for study eligibility? For patients who are randomised to the control arm who have had benefit from first-line FOLFIRINOX, will they continue on the same regimen or must they change to a different second-line regimen? In Table 1, systemic chemotherapy spans from 3 months post-allocation to 12 months. This seems to suggest that patients take a 3-month treatment break between randomisation and continuing chemotherapy, whereas the protocol states that they should receive chemotherapy within 4 weeks after randomisation.
--

	 3. At what point do patients receive chemotherapy after IRE? Will this occur in the event of disease progression, or irrespective of tumour response to the procedure? 4. The document titled SEPTRE_checklist_LAP-PIE appears to be a copy of the protocol. Minor comments  1. Please clarify in the manuscript how many IRE treatments will be delivered. The abstract states that only one procedure will be delivered, but this is not clear in the protocol. 2. P8: The authors should update the recommendations for the systemic treatment of unresectable and metastatic pancreatic cancer, as they currently do not align with NICE guidelines. 3. P10: capecitabine and gemcitabine is not a standard second-line treatment for advanced pancreatic cancer, as per NICE guidelines. The URL for the reference given for this treatment (reference 18) is not accessible. 4. Please change CKD 3 to CKD 3 Stage 3 and state the criteria for this. 5. Why are neurological conditions an exclusion criterion? 6. P18: what is the radiological definition for successful treatment of the entire tumour? 7. P19: there seem to be 2 definitions for overall survival (time from diagnosis and time from randomisation). Survival is typically measured from randomisation. 8. P19: please explain how progression free survival will be measured as a binary outcome, as it is typically a time-to-event outcome. 9. Figure 1: unresectable, locally advanced pancreatic cancer should be included as a key inclusion criterion.
--	---

VERSION 1 – AUTHOR RESPONSE

Reviewer 1:

General remarks:

The feasibility of percutaneous IRE in LAPC patients has been shown in multiple publications. Why would the authors go through the effort of designing and conducting a randomized controlled trial and not power it for efficacy?

We would agree that the technical feasibility of percutaneous Irreversible Electroporation (IRE) in Locally Advanced Pancreatic Cancer (LAPC) patients has

been demonstrated in previous publications. However, we wish to demonstrate whether IRE confers survival benefit and for this a randomised controlled trial is required. To randomise patients requires all aspects of feasibility to be explored including recruitment of patients to this type of study, adherence to treatment protocols and preliminary indication of efficacy. With this information the first trial sufficiently powered to demonstrate efficacy can be designed and will hopefully follow this study.

The introduction/background section is hard to read. The authors are encouraged to rethink the structure of this section. A structure could be to first explain the standard treatment program and then go on to explain how IRE has been implemented in the existing clinical trials. Finally, please explain why the implementation in the current study is chosen and what the aims of the study are.

Thank you. The introduction section has been re-formatted to include the reviewer's suggestions on structure.

The reason behind the implementation of the LAP-PIE trial is clearly stated in the last paragraph of the Introduction and states that for a larger, appropriately designed RCT evaluating the efficiency and cost effectiveness of IRE in LAPC, there first needs to be a feasibility trial, this is further expanded in the Discussion section that demonstrates the justification of such a trial. Finally, the authors have chosen not to include the aims in this section in keeping with the guidelines from BMJ. In accordance with the BMJ guidelines the aims of the study are included under the section title 'Objectives'.

Are IRE treatments being performed in all centres or just one. Please elaborate on the prior experience with IRE treatment.

IRE will be performed in all centres registered to take part in the trial. The centres taking part in the LAP-PIE trial will be tertiary specialist pancreatic centres. The Interventional Radiologists (IR) delivering the procedure in the trial will be mandated to have performed 5 previous pancreas or liver IRE procedures as a minimum prior to delivering IRE to participants in the trial. There will be a system of mentoring from experienced IRE interventional radiologists. Additional technical support will be available from AngioDynamics®, the manufacturers of IRE. This has been clarified in the protocol, please see page 10 lines 9-13

How is safety maintained throughout the study? The protocol does not appear to contain any constraints or limitations on how many and how severe adverse event are acceptable to continue the trial. Please consider formulating a clear guideline for when the trial is considered too dangerous to continue.

The safety of participants is an important aspect and considering the Reviewer's comments, the authors have added a paragraph clarifying the safety guidelines.

Please see page 20 lines 3-8.

Specific remarks:

Introduction/background

Page 8, line 51-59: The authors wrote that patients not eligible for FOLFIRINOX are treated with gemcitabine monotherapy and then goes on to say that NICE guidelines recommend GemCap. This seems to be inconsistent. The authors are encouraged to rephrase or explain?

This point has been clarified following the Reviewer's comment to explain that NICE guidelines recommend FOLFIRINOX in patients who are able to tolerate this regime and in those unfit to tolerate this, Gemcitabine based therapy is recommended.

Please see page 6 line 14.

Page 9, line 11-12: The headline is not consistent with the content of the paragraph. Upfront chemotherapy, consolidative chemotherapy or downstaging chemotherapy are better descriptors than "First-line chemotherapy" in this context. Maybe "timing of

IRE treatment” is a better headline.

Accepted. The headline has been changed to include the word ‘Upfront’. Please see page 7, line 15.

Page 9, line 19-22: The authors wrote that “studies” report an OS of 27 months, but only reference one study. The authors are encouraged to rephrase.

Accepted. The paragraph clarifies that the Overall Survival (OS) refers to a systematic review (page 7 line 18-19)

Page 9, line 38-48: Considering that the cited study is a systematic review, the statement “and in this cohort” seem out of place. The authors are encouraged to rephrase.

Accepted. The statement ‘and in this cohort’ has been removed

Page 10, line 3-6: Please refer to the specific guidelines use to define LAPC. In this case I think it is the NCCN guideline, as these are referenced in the paragraph.

Accepted. The definition of LAPC as per NCCN has been included and referenced. (Page 8 line 7-10)

Page 10, line 8-27: This should be in the methods section.

Accepted. The sentence, “If new agents for second line therapy are approved by NICE prior to or during the LAP-PIE trial these can be offered in the second line chemotherapy arm and following IRE therapy if considered appropriate by the treating oncologist.” has been moved to the ‘Methods’ section as per Reviewer’s suggestion (page 10, line 3-5)

Page 11, line 15: “during surgery or electively in the IR suite”. I believe that the authors meant to differentiate between open surgery (open in-situ IRE) and percutaneous image-guided IRE. Please make this clearer. Also, IR (interventional radiology?) is not explained.

The authors have clarified that IR stands for Interventional Radiology (please see page 8 line 22.) As per the reviewer’s suggestion, the authors have clarified what is meant by open in situ IRE i.e., IRE delivered through a mid-line laparotomy or laparoscopically (see page 8 line 21) and percutaneous IRE (see page 8 line 22)

Page 11, line 22-24: It is not completely true that no data exist regarding the number of treatments performed and outcomes. Please see: Philips P, Hays D, Martin RCG (2013) Irreversible Electroporation Ablation (IRE) of Unresectable Soft Tissue Tumours: Learning Curve Evaluation in the First 150 Patients Treated. PLoS ONE 8(11): e76260. doi: 10.1371/journal.pone.0076260>

Please see page 9 line 1 which clarifies that the information regarding the number of IRE treatments performed by individual clinicians and that the subsequent impact on patient outcomes remains poorly understood and needs further clarification.

Page 11, line 30-37: Please reread the article. The authors find that QoL is impacted in the short term but suggest that this could be the results of other interrelated factors.

The wording has been clarified as per the Reviewers suggestion, please see page 9 lines 6-7.

Page 11, line 36-39: What is meant by “no recognised QoL scores that compare IRE with no treatment or systemic chemotherapy”? I suggest rephrasing to “We were unable to find any studies comparing the QoL changes after IRE to a control group.”

Accepted. Please see page 9, lines 7-9.

Page 11, line 39-44: None of the published data, to date, has found any positive effects on QoL. What data are the authors referring to?

Information on the impact of QoL of patients undergoing IRE therapy for pancreatic cancer is limited in the literature. This is the point the authors wished to make and has been clarified in the manuscript following the Reviewer’s comment. Kindly refer to page 9 line 8-9.

Page 11, line 46-52: This paragraph seems out of place and should be removed. This paragraph explains why the LAP-PIE trial is needed and the anticipated next steps following on from trial completion. It is therefore appropriate to place it at the end of the introduction section. The authors would argue that it is an essential paragraph that summarises the gaps in the knowledge regarding IRE in LAPC and introduces the LAP-PIE study as a feasibility trial aimed to facilitate a larger RCT.

Page 11, line 57-13: Why demonstrate cost-effectiveness, when efficacy has not been established? Please explain.

The trial's primary objective is to assess feasibility of recruitment into a trial of this nature. It is not designed to assess cost-effectiveness. The cost of related health resources in both arms will be evaluated as secondary objectives. It is hoped that the information regarding health costs can be used to guide a larger RCT which would evaluate efficacy and cost effectiveness.

Also, if the aim of the trial is to assess "whether they would accept randomization to IRE or standard of care chemotherapy, whether they would comply with treatment and study data collection", why not set up an even smaller trial to assess this prior to a larger randomized controlled trial? Considering that IRE has been linked to severe morbidity and mortality, please explain why 50 patients are needed to assess this!

The primary aim of the LAP-PIE trial is feasibility of recruitment to a trial of this design. A sample size of less than 50 patients would not adequately be able to assess the feasibility of recruiting patients with LAPC.

Methods section:

Inclusion criteria: Tissue confirmation of what? Adenocarcinoma? Please be specific.

Accepted. Please see revised Figure 2 which specifies that one of the Inclusion criteria is tissue confirmation of adenocarcinoma.

What defines a "successful completion of FOLFIRINOX"? Please elaborate.

Patients who have completed between 3- to 6-months of FOLFIRINOX will be deemed to have successfully completed FOLFIRINOX treatment for this trial. This is clarified on page 10, line 1. This duration has been specifically chosen to identify patients who progress or develop early metastases and prevent cumulative toxicity of prolonged chemotherapy prior to IRE. The authors would like to make the point that the LAP-PIE study is a trial that aims to better understand the potential impact of adding IRE to the current treatment available for patients with LAPC. As such the protocol mandates a clear structure of the intervention (IRE). The standard of care, whilst an important aspect of overall clinical care, is not being studied by this trial. Therefore, it is appropriate to mandate a range in the treatment duration of FOLFIRINOX to strike a balance between acceptability to recruiting centres and avoiding a greater than necessary variability in the protocol.

Consider rephrasing inoperable to unresectable

Accepted. 'Inoperable' has been replaced with 'unresectable'.

Please explain "IR".

IR stands for Interventional Radiologist. This has been explained in the legend of figure 2.

Exclusion criteria: What if FOLFIRINOX has been given in the second line but prior to inclusion? It is seldom, but can happen in some patients, because of positive changes in performance status after initial therapy. Again, please be precise in the use of the words first/second-line chemotherapy. Also, this criterion seems to be redundant as patients are only included if they have completed FOLFIRINOX.

If FOLFIRINOX is not the first chemotherapy agent completed by patients, then they are not eligible for the trial. This has been clarified following the Reviewer's comment, please refer to page 10, line 18-20.

Please consider reading the literature on IRE ablations in the present of metallic

biliary stents and consider adding this as an exclusion criterion.

A metallic stent has not been considered to as an exclusion criterion based on previous published experimental data that suggests that IRE in the presence of a metallic stent is safe. 'The influence of a metal stent on the distribution of thermal energy during irreversible electroporation' by Scheffer et al PMID 26844550, in which the authors found that there was no temperature difference, in vivo between ablations with or without a metal stent.

Page 19, line 12-22: The authors have previously written that both groups will be offered addition chemotherapy (please consider calling it addition palliative chemotherapy or adjuvant chemotherapy). In this paragraph, the author writes that patients are randomized to receive chemotherapy. Please be consistent. Consider rephrasing to "chemotherapy only".

Accepted. The wording has been amended to include 'chemotherapy alone'. Please see page 16, line 21.

Outcomes: Why is cost-effectiveness not mentioned here, when this was considered a primary aim of the trial in the background section?

Cost effectiveness is not a primary aim in the trial. The primary aim for this study is feasibility of recruitment which is clearly stated under the 'Outcomes' section and the 'Introduction' section.

In the secondary outcomes, please be specific on what is measured, e.g., "...safety and complications of IRE treatment, technical success of intervention..." is not apparently measurable outcomes

The paragraph on secondary aims has been expanded and clarifies the measurable outcomes that will be evaluated as per the reviewer's suggestion. Please see page 17 lines 8-18 and lines 18-21.

Also, how is PFS "measured as a binary outcome"? Please explain.

This has been amended, please see page 17, line 17.

How are QoL evaluated?

Quality of life will be assessed through the validated QoL questionnaire (EQ5D-5L).

This is explained on page 18 line 1.

Are the authors interested in measuring changes in certain QoL outcomes or just means at specific points in the follow-up-period?

The authors are interested in the impact, if any, of all outcomes as assessed by the validated QoL questionnaire.

Sample size: Please read comment above.

The trial has been designed to assess feasibility of recruitment of LAPC patients.

This is based on the number of patients presenting to tertiary pancreas centres in the UK. The reasoning behind selecting 50 patients is explained on page 19 lines 4-15 under the section titled 'Assessment of feasibility'.

Page 20, line 52: Interpretation of results should be objective!

The authors feel very strongly about patient and public involvement (PPI) in the LAPPIE trial. This is reflected in the robust PPI involvement from the early stages of the trial, and it is hoped that this involvement will continue throughout the length of the trial and beyond. It is insensitive and erroneous to suggest that patients and patient support groups are incapable of interpreting results in a meaningful and objective manner.

Assessment of feasibility: The evaluated recruitment rate seems too high, considering the number of patients and the selection criteria. The authors should ask themselves how many patients are still in PS 0/1 after FOLFIRINOX treatment, are unresectable and have tumours below 3.5cm in diameter.

The recruitment rate is appropriate for the number of unresectable cases of locally advanced pancreatic cancer in the United Kingdom. The average number of patients

presenting with unresectable LAPC to specialist, tertiary pancreas centres in the UK is 60-65 patients a year (Royal Free Hospital Audit data). Following chemotherapy with FOLFIRINOX, 65% of patients will remain with unresectable LAPC, this equates to approximately 40 patients a year per centre. There a total of 5 centres participating in the trial with a potential to include more centres if recruitment rate halfway through the trial is deemed to be low. This information is explained on page 19, lines 5-14). Based on this, the authors feel that the anticipated recruitment rate is appropriate and achievable for the trial.

Assessment of complications: Consider using the same classification system in both groups of patients. Otherwise, they will not be comparable in the analysis.

Additionally, what is the point in comparing SAEs to the reference and not to the other group? Please explain why this is not compared between groups.

It is not appropriate to use the same adverse event classifications to compare in both groups as IRE is an invasive procedure. Therefore, complications associated with IRE will be measured against an appropriate and validated grading system for invasive procedures, namely the Clavien Dindo Classification. Similarly, complications associated with the chemotherapy alone arm will be measured against the specific drug's reference safety information (RSI) within the current summary of product characteristics (SmPC) for that drug. It is unlikely that complications from the different arms will be the same as the treatments are vastly different. The rates of complications in each arm are a far superior objective to analyse and rates of complications between groups will be measurable and applicable even if different classification systems have been used.

Translational study: Please be specific about what exactly is measured.

The exact immune response markers have been clarified. Please see page 20 line 25 and page 21 line 1.

Ethics:

Please reference the specific REC.

The specific Research Ethics Committee and the approval reference are now included. Please see page 21, line 13-14.

Please register protocol in a public database e.g., clinicaltrials.gov.

The trial has been registered with the International Standard Randomised Controlled Trial Number (ISRCTN) <https://doi.org/10.1186/ISRCTN14986389>. This information has been added to the Ethics section please see page 21 line 13 to 16.

Discussion:

Page 24, line 10-15: Propensity score matched studies compare two groups of patients, while single-arm follow-up studies does not. Consider reporting the difference between the two groups instead of the OS of one group.

This has been amended to allow for comparison between the overall survival rates. Please see page 22 line 14.

Page 25, line 56-3: The authors should consider the context of the treatment here.

FOLFIRINOX has been shown to reduce tumours to a resectable size, even when this is not evident from the clinical images. Please read: "Ann Surg Oncol (2015) 22:1153–1159 DOI 10.1245/s10434-014-4225-1". The authors are encouraged to consider how they will avoid including patients that are potentially resectable.

The authors agree that FOLFIRINOX has the potential to downgrade tumours. To ensure that the patients included in trial do not have potentially resectable disease, the protocol mandates a full and thorough specialist multi-disciplinary (sMDT) review, which will include both surgeons as well as radiologists, following first-line chemotherapy and prior to consideration in the trial. Furthermore, the protocol has a clear consensus and definition of what constitutes as 'locally advanced pancreatic cancer' as per NCCN guidelines.

Page 26, line 40-43: This information should be mentioned in the methods section. Please explain when the blood samples are taken and how the comparison of the results will be performed.

Translational studies are not mandated in the study protocol and therefore not included in the methods section. Blood tests for the translational nested study will be taken at baseline during the screening visit, 6 hours following IRE and 6 days after the IRE procedure. This has been explained in the section describing nested substudies, under the title 'Translational studies.

Page 26, line 47-52: "analysing...efficacy". What is meant by "defined" this sentence? Please rephrase.

This has been rephrased to provide clarity. Please see page 26 line 6

Page 27, line 12-15: Please provide reference.

Please see references 14-16, 21-23 and 34 which have been added as per Reviewer's suggestion.

Page 27, line 38-41: "The LAP-PIE trial will provide evidence for supporting or refuting the role of IRE in patients with locally advanced pancreatic cancer." How is this possible if the study is not powered to do so?

This sentence has been removed.

Reviewer 2

Major comments

This feasibility study is described as the pre-cursor to conducting a larger trial that will assess the cost-effectiveness of this procedure. If a feasibility trial is positive, typically the next step would be to conduct a randomised controlled trial to demonstrate efficacy. Cost- effectiveness becomes a relevant question only after efficacy is demonstrated in a randomised trial, so it unclear why a cost-effectiveness study would directly follow a positive feasibility trial.

Indicative procedure related costings will be collected in the feasibility study which will inform the cost evaluation in the subsequent study on efficacy.

More details should be provided regarding the systemic treatment components of the study, as they are likely to affect tolerability of the overall protocol and potentially survival outcomes. How many cycles of FOLFIRINOX will patients receive prior to restaging and being assessed for study eligibility? For patients who are randomised to the control arm who have had benefit from first line FOLFIRINOX, will they continue the same regimen, or must they change to a different second-line regimen? Patients will receive between 3-6-months of FOLFIRINOX prior to restaging and being assessed for study eligibility. It is hoped that this will allow for the identification of those patients who progress or develop metastases early, and therefore would not be considered for the trial, but also prevent cumulative toxicity caused by sustained and prolonged FOLFIRINOX treatment. The second line chemotherapy regime will be selected at the discretion of the treating oncologist and will adhere to NICE guidelines and local protocols. Patients randomised to the control arm who have benefited from first line FOLFIRINOX may continue the same regimen if this is deemed to be appropriate by their treating oncologist. The protocol does not mandate a switch in chemotherapy regime in either arm of the study.

In Table 1, systemic chemotherapy spans from 3 months post-allocation to 12 months. This seems to suggest that patients take a 3-month treatment break between randomisation and continuing chemotherapy, whereas the protocol states that they should receive chemotherapy within 4 weeks after randomisation.

Table 1 summarises the overall timetable of the LAP-PIE trial. 3-months through to 12-months refers to the time points at which patients will be followed-up and not the time at which they receive chemotherapy. Nonetheless, following Reviewer's comments, Table 1 has been amended to make this clearer. Kindly refer to Table 1

version ii.

At what point do patients receive chemotherapy after IRE? Will this occur in the event of disease progression, or irrespective of tumour response to the procedure? Patients randomised to receive IRE will receive chemotherapy irrespective of disease progression. The timing of IRE and choice of chemotherapy agent will be decided by the treating oncologist. The LAP-PIE trial is a pragmatic study that aims to understand the effect of adding IRE to standard of care, chemotherapy. This point has been clarified. Please see page 16, line 12-14.

The document titled SEPTRE_checklist_LAP-PIE appears to be a copy of the protocol.

The SEPTRE checklist has been amended. Please refer to version II.

Minor comments

Please clarify in the manuscript how many IRE treatments will be delivered. The abstract states that only one procedure will be delivered, but this is not clear in the protocol.

Patients randomised to receive IRE will have one procedure. This has been clarified in the manuscript, please see page 10, lines 2-3.

P8: The authors should update the recommendations for the systemic treatment of unresectable and metastatic pancreatic cancer, as they currently do not align with NICE guidelines.

Accepted. The recommendations for systemic treatment have been amended. Kindly refer to page 6 line 8-17

P10: capecitabine and gemcitabine is not a standard second-line treatment for advanced pancreatic cancer, as per NICE guidelines. The URL for the reference given for this treatment (reference 18) is not accessible.

Reference 18 has been removed as the URL is no longer accessible. Second-line chemotherapy for pancreatic cancer has been updated.

Please change CKD 3 to CKD 3 Stage 3 and state the criteria for this.

Accepted. This has now been changed to 'CKD Stage 3' with the criteria stated.

Please see page 11 line 2-3.

Why are neurological conditions an exclusion criterion?

Epilepsy and other neurological conditions have been excluded due to safety concerns and following advice from manufacturers of the IRE generator. Whilst the impact of IRE on pulsatile brain stimulation is still being investigated, the authors feel that excluding patients with neurological conditions is a safe precaution. This has been clarified on page 11 lines 6-10.

P18: what is the radiological definition for successful treatment of the entire tumour?

Intra-operative images taken during IRE procedure will be reviewed by site radiologists, if the entire tumour has been treated based on these images, the procedure will be deemed as successful. This is clarified on page 16, lines 8-13.

P19: there seem to be 2 definitions for overall survival (time from diagnosis and time from randomisation). Survival is typically measured from randomisation.

Overall survival will be measured from randomisation. This has been clarified following Reviewer's comment. Please see page 17, line 17.

P19: please explain how progression free survival will be measured as a binary outcome, as it is typically a time-to-event outcome.

Progression free survival will be measured as a time-to-event outcome, and this has been clarified. Please see page 17, line 17.

Figure 1: unresectable, locally advanced pancreatic cancer should be included as a key inclusion criterion.

Locally advanced pancreatic cancer has been added to the inclusion criteria. Please see revised Figure 2.

VERSION 2 – REVIEW

REVIEWER	Flak, Rasmus Virenfeldt Aalborg Universitetshospital, Department of Gastrointestinal Surgery
REVIEW RETURNED	24-Aug-2021

GENERAL COMMENTS	Thank you for letting me review the revised manuscript. Most of my prior remarks have been addressed. The answers are clearly stated, and the manuscript have been corrected accordingly. Based on reading the revised manuscript I have a few minor remarks: 1) Page 3, Line 18: “This trial is the first...” Unfortunately this is no longer true as the first RCT has recently been published. See https://dx.doi.org/10.12998/wjcc.v8.i22.5564 2) Page 4, Line 20: “The protocol has received a favorable opinion...” The phrasing makes me unsure whether the protocol has been approved by the REC or not. I can read below that the protocol has been approved. Please consider rephrasing. 3) Page 9, Line 11-19: Please see first remark. The study cited above compared OS in gemcitabine + IRE treated patients versus gemcitabine only in a randomized controlled study. They find a clear survival advantage. However, in the LAP-PIE trial you treat patients that have been treated with FOLFIRINOX, which is a clear recommendation by NICE and other organization. Thus, you could make it clear in this section, that evidence does exist for the efficacy of IRE, but your trial will evaluate whether this is true in the modern context of FOLFIRINOX treated patients. 4) Page 14, Line 9-14: Consider using a standardized approach for energy delivery. This is not a problem with the manuscript, but it would strengthen the trial design. 5) Page 15, Line 1-3: Local control according to RECIST is associated with some problems as the lesion may appear larger on follow-up scans due to edema and scar formation. Consider reading https://doi.org/10.3390/cancers13122862. Again, this is not a problem with the manuscript, but an issue that should ideally be taken into consideration. 6) Page 20, Line 19-21: As mentioned above in 1) & 3).
---

VERSION 2 – AUTHOR RESPONSE

'The 'Strengths and Limitations' section has been revised to include short bullet points relating specifically to the methodology of the trial. Please see page 3 lines 18-23 and page 4 lines 1-5.'

Please make sure it is clear in your ethics statement whether the ethics committee has approved the study. The current wording is unclear.

'The wording for ethics approval has been changed to make it clearer that the study has been granted approval. See page 4 line 22.'

'This feasibility study is described as the pre-cursor to conducting a larger trial that will assess the cost-effectiveness of this procedure. If a feasibility trial is positive, typically the next step would be to conduct a randomised controlled trial to demonstrate efficacy. Cost-effectiveness becomes a relevant question only after efficacy is demonstrated in a randomised trial, so it unclear why a cost effectiveness study would directly follow a positive feasibility trial.'

The justification of using the feasibility trial to help design a larger trial has been clarified. See page 4 line 18-20. This feasibility trial will be used to inform a larger RCT powered to evaluate efficacy. This study will analyse indicative procedure related costs to inform the cost evaluation aspect of a subsequent larger trial.

'More details should be provided regarding the systemic treatment components of the study, as they are likely to affect tolerability of the overall protocol and potentially survival outcomes. How many cycles of FOLFIRINOX will patients receive prior to restaging and being assessed for study eligibility? For patients who are randomised to the control arm who have had benefit from first line FOLFIRINOX, will they continue the same regimen, or must they change to a different second-line regimen?'

Editor: Please ensure that this information is included in your revised manuscript and state in your response letter where the revisions have been made in the text.'

Further clarification on the use of FOLFIRINOX in this study has been made (see page 10 line 3-4 and line 9-11). The revision clarifies that patients will receive 3-6-month duration of FOLFIRINOX prior to being randomised and that patients randomised to the control arm may continue with FOLFIRINOX if this is deemed appropriate by the treating oncologist. The LAP-PIE trial does not mandate a switch in chemotherapy in either arm of the study.

'Page 3, Line 18: "This trial is the first..." Unfortunately this is no longer true as the first RCT has recently been published.'

The trial that Reviewer 1 mentions has been included in the references and is referenced in the additions made in the text (see page 3 line 18-19 and reference number 25)

'Page 4, Line 20: "The protocol has received a favorable opinion..." The phrasing makes me unsure whether the protocol has been approved by the REC or not. I can read below that the protocol has been approved. Please consider rephrasing.'

The protocol has been approved by London-Brent Research Ethics Committee, reference number 21/LO/0077. The wording has been amended to clarify this. See page 4 line 22.

'Page 9, Line 11-19: Please see first remark. The study cited above compared OS in gemcitabine + IRE treated patients versus gemcitabine only in a randomized controlled study. They find a clear survival advantage. However, in the LAP-PIE trial you treat patients that have been treated with FOLFIRINOX, which is a clear recommendation by NICE and other organization. Thus, you could make it clear in this section, that evidence does exist for the efficacy of IRE, but your trial will evaluate whether this is true in the modern context of FOLFIRINOX treated patients.'

The use of IRE in a randomised controlled trial with FOLFIRINOX has not been reported and therefore this feasibility trial is novel. The study mentioned is an important study as it is the first RCT in IRE as Reviewer 1 correctly points out. This has been referenced in the manuscript and we have made it clear that the LAP-PIE study focuses on the effect of IRE in LAPC patients treated with FOLFIRINOX; a more modern chemotherapy agent and one that is currently recommended by NICE (see page 9 line 15-20).

'Page 14, Line 9-14: Consider using a standardized approach for energy delivery. This is not a problem with the manuscript, but it would strengthen the trial design.'

Unlike liver IRE, pancreas IRE has no consensus on a standard treatment protocol. To mitigate this variation, the protocol defines a range of IRE treatment parameters (see page 14, line 17-24).

'Page 15, Line 1-3: Local control according to RECIST is associated with some problems as the lesion may appear larger on follow-up scans due to edema and scar formation. Consider reading. Again, this is not a problem with the manuscript, but an issue that should ideally be taken into consideration.'

We agree with this comment. The protocol mandates that imaging be reviewed twice, centrally by an experienced interventional radiologist and by site radiology leads. See page 15 line 16-17.

'Page 20, Line 19-21: As mentioned above in 1) & 3).'

The novelty of this study remains, as it will be the first RCT to report of IRE in addition to FOLFIRINOX in LAPC patients. The study by Ma et al reporting on IRE and concurrent gemcitabine has been included and referenced (see page 21 line 23-24), however the publication of this study does not diminish or lessen the impact of the LAP-PIE trial.

Thank you for your consideration of this manuscript.

Sincerely,

ZL Rai and BR Davidson